

# Exploring spatial nonstationary environmental effects on Yellow Perch distribution in Lake Erie

Changdong Liu[1], Junchao Liu[1], Yan Jiao[2], Yanli Tang[1] and Kevin B. Reid[3]

[1] Department of Fisheries, Ocean University of China, Qingdao, Shandong, China
[2] Department of Fish and Wildlife Conservation, Virginia Polytechnic Institute & State University, Blacksburg, VA, USA
[3] Department of Integrative Biology, University of Guelph, Guelph, ON, Canada

## ABSTRACT

**Background:** Global regression models under an implicit assumption of spatial stationarity were commonly applied to estimate the environmental effects on aquatic species distribution. However, the relationships between species distribution and environmental variables may change among spatial locations, especially at large spatial scales with complicated habitat. Local regression models are appropriate supplementary tools to explore species-environment relationships at finer scales.
**Method:** We applied geographically weighted regression (GWR) models on Yellow Perch in Lake Erie to estimate spatially-varying environmental effects on the presence probabilities of this species. Outputs from GWR were compared with those from generalized additive models (GAMs) in exploring the Yellow Perch distribution. Local regression coefficients from the GWR were mapped to visualize spatially-varying species-environment relationships. *K*-means cluster analyses based on the *t*-values of GWR local regression coefficients were used to characterize the distinct zones of ecological relationships.
**Results:** Geographically weighted regression resulted in a significant improvement over the GAM in goodness-of-fit and accuracy of model prediction. Results from the GWR revealed the magnitude and direction of environmental effects on Yellow Perch distribution changed among spatial locations. Consistent species-environment relationships were found in the west and east basins for adults. The different kinds of species-environment relationships found in the central management unit (MU) implied the variation of relationships at a scale finer than the MU.
**Conclusions:** This study draws attention to the importance of accounting for spatial nonstationarity in exploring species-environment relationships. The GWR results can provide support for identification of unique stocks and potential refinement of the current jurisdictional MU structure toward more ecologically relevant MUs for the sustainable management of Yellow Perch in Lake Erie.

Corresponding author
Yanli Tang,
changdong_ouc@163.com

## INTRODUCTION

Estimating the key relationships between species distribution and environmental variables is essential for natural resource conservation and ecosystem-based fishery management (*Grüss et al., 2017*). A large number of published papers reported that environmental variations caused changes in species abundance and distribution (*Tseng et al., 2013*; *Barbeaux & Holloweb, 2017*; *Liu et al., 2017*; *Muška et al., 2018*). Many biotic and abiotic factors, as well as the interactions among them, drive the species distribution at variable spatial-temporal scales. It is challenging to disentangle environmental effects on species distribution because of the spatial dynamic response of species to environmental variation. Exploring environmental effects on species distribution at a single large spatial scale may mask the intrinsic relationships between them at finer scales. Accounting for spatial nonstationarity can improve our understanding of the interactive process between species distribution and environmental variables at various spatial scales (*Windle et al., 2010*, *2012*; *Sadorus et al., 2014*; *Liu et al., 2017*; *Li et al., 2018*; *Bi et al., 2019*).

Global regression models are the predominant methods used to estimate environmental effects on species distribution presently. Generalized additive models (GAMs) are also popular and are considered superior to generalized linear models (GLMs) for estimating the nonlinear relationships between species distribution and environmental variables (*Canepa et al., 2017*; *Grieve, Hare & Saba, 2017*; *Hemami et al., 2018*). These models estimate one average regression parameter, independent of locations and directions, for each explanatory variable over the entire study area. Due to the complexity of aquatic ecosystems and the dynamic interaction between biology and environment, the assumption of spatial stationary relationships between biological and environmental factors may be violated, especially at large spatial scales. Thus, local regression models can be effectively complement global models in inferring species-environment relationships at finer scales (*Fotheringham, Brunsdon & Charlton, 2002*).

The spatial expansion model (SEM) is one of the early methods used to estimate spatially-varying ecological relationships (*Fotheringham, Charlton & Brunsdon, 1997*; *Fotheringham, Charlton & Brunsdon, 1998*; *Li, Jiao & Browder, 2016a*, *2016b*). In SEM, each regression parameter itself is a function of spatial location and the form of the function (e.g., linear, polynomial) is determined by prior knowledge. The results from SEM are sensitive to the spatial expansion function and are hard to explain due to use of the complex function (e.g., high-order polynomial). The geographically weighted regression (GWR) model is a natural evolution of SEM. In GWR, a local regression model is fitted at each sample location using neighborhood observations. The weights of neighborhood observations on regression points are defined according to spatial dependence and the weighted least square method can be used to fit a local regression model. The main advantage of GWR is that it yields a set of parameter estimates at each sample location and the regression coefficients for each environmental variable can be mapped over the study area to visualize spatially-varying ecological relationships (*Fotheringham, Brunsdon & Charlton, 2002*).

Lake Erie is the smallest in volume but biologically most productive of the Laurentian Great Lakes (*Hartman, 1972*). The lake is separated into west, central and east basins

as the significant environmental differences among them. Yellow Perch (*Perca flavescens*) is one of the most important commercial and recreational species in Lake Erie and plays an important role in regional economic development (*YPTG (Yellow Perch Task Group), 2015*). Based on morphology, Lake Erie is comprised of three main basins. Management of this species is based on management units (MUs), which correspond coarsely to the three basins, with the larger central basin separated into two MUs (*YPTG (Yellow Perch Task Group), 2015*). The current division of four MUs for Yellow Perch fisheries management is based on limnology, geography and tagging studies but also on historical statistical harvest district boundaries (*YPTG (Yellow Perch Task Group), 1983*; *Cowan & Paine, 1997*). The definitions of MU boundaries are convenient for landing and reports of harvesting, and are jurisdictional in nature, with uncertain ecological relevance. Genetic and morphological studies have provided evidence of differences in Yellow Perch stocks among MUs (*Kocovsky & Knight, 2012*; *Sepulveda-Villet & Stepien, 2011*); however, recent tagging studies indicated large scale fish movement among MUs especially among MUs 2 and 3 (*Peterson, 2014*; *Personal communication with Andy Cook from Ontario Ministry of Natural Resources*). We expect the discrete stocks will, to some degree, present differential responses to environment changes. Hence exploring spatially-varying species-environment relationships can provide support for identification of distinct stocks and refinement of MUs.

Water temperature, water depth, water transparency and dissolved oxygen (DO) are the key habitat variables to affect Yellow Perch distribution and several published studies had applied global regression models to estimate the environmental effects on Yellow Perch distribution (*Power & Heuvel, 1999*; *Arend et al., 2011*; *Bacheler, Paoli & Schacht, 2011*; *Yu, Jiao & Winter, 2011*; *Manning et al., 2013*; *Liu et al., 2018*). In this study, we questioned the spatial-stationarity assumption and explored spatially-varying species-environment relationships of Yellow Perch in Lake Erie using the GWR method. The performance of GWR was compared with that of GAM to test whether GWR outperforms GAM. We mapped the coefficient estimates of GWR to visualize ecological relationships and characterized the special zones of these relationships. Our objective was to discover species-environment relationships at finer scales to support possible refinement of the MUs toward improved management based on discrete Yellow Perch stocks.

## MATERIALS AND METHODS

### Study area and data sources

The partnership index survey (PIS) was conducted using the standard gillnets by the Ontario Ministry of Natural Resources and Forestry and the Ontario Commercial Fisheries' Association in the Canadian waters of Lake Erie in annual late summer and early fall. A depth-based stratified random sampling design was used and the number of sample sites was determined according to the surface area of each depth stratum (see *Berger, Jones & Zhao, 2012*; *Pandit et al., 2013*; *Liu et al., 2018*). The survey gillnet gangs composed of 14 different mesh sizes were set on the bottom and suspended (canned) in the water column with a mean soak time of 20 h.

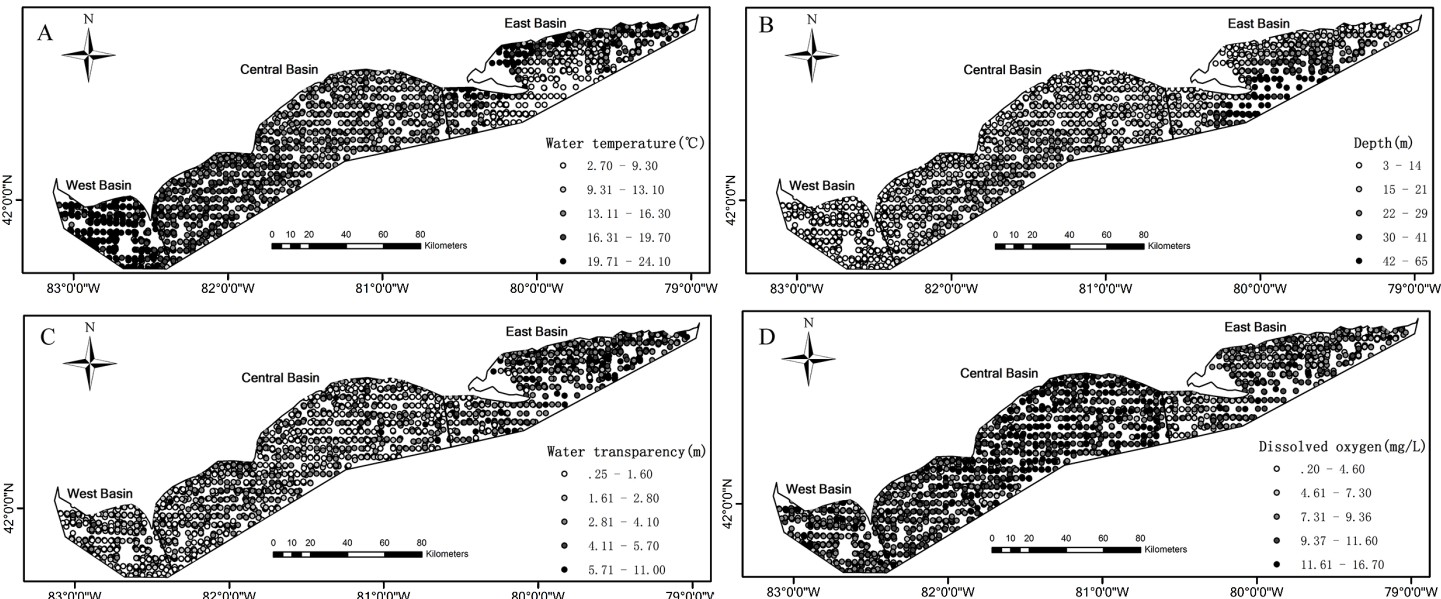

**Figure 1** Spatial distributions of the measured values for environmental variables based on the partnership index survey (PIS) in the Canadian side of Lake Erie from 1989 to 2015. (A) Water temperature. (B) Water depth. (C) Water transparency. (D) Dissolved oxygen. The bold black lines in Lake Erie are the separate lines among the three basins.                               

Partnership index survey measured the catch in weight and in number of Yellow Perch by age (age 0, age 1, age 2, etc.) at each sample. The ages of fish caught were estimated, initially by scales, and later by otoliths (*YPTG (Yellow Perch Task Group), 2015*). The environmental effects on Yellow Perch distribution depend on life stages (*Liu et al., 2018*). Accordingly, we divided the catches into juveniles (age < 2) and adults with (age ≥ 2) because the age of recruitment to the Yellow Perch fishery was defined as age-2 fish. The catches for juveniles were simplified as 0/1 indicating absence/presence of fish at each sample. The corresponding operation was done for adults. Water temperature (°C), water transparency (m) and DO concentration (mg/L) were measured at the depth of gillnet (the depth from the water surface to the bottom of gillnet). Water transparency was estimated based on the visual distance of Secchi disk. To calculate distance, sample location coordinates denoted as longitude and latitude were converted to plane coordinates by the North American Datum 1983 Universal Transverse Mercator 17N projection. As benthivorous fish, only 6% in weight of the total Yellow Perch catch was found in the canned gillnets, and therefore we analyzed data from bottom gillnets only. The survey data from 1989 to 2015 included a total of 2,502 samples that were analyzed in this study after removing the missing/erroneous values (108) in the catch or environmental data (Fig. 1).

## Model development

The fish catch data for each sample were simplified as 0/1 to indicate absence/presence of Yellow Perch. We built models to estimate the relationships between the presence probability of Yellow Perch and environmental factors. Water temperature, water depth,

water transparency and DO concentration were used as explanatory variables in the model analysis because they were surveyed contemporaneously with the fish data and were proved to be the key habitat variables to Yellow Perch distribution (*Liu et al., 2018*). A preliminary variance inflation factor (VIF) analysis was conducted to test for multicollinearity of explanatory variables (*Zuur, Ieno & Elphick, 2010*). The environmental factors with VIFs greater than three were excluded in the next model analysis (*Sagarese et al., 2014*). As all the VIFs were less than two, the four environmental variables were included in the following model analysis.

We first applied GAMs to estimate the environmental effects on the presence probabilities of juveniles and adults. GAMs extend the GLMs by replacing the linear predictors with spline functions to estimate the nonlinear relationships between response and explanatory variables (*Wood, 2006*). In the study, GAMs are denoted as:

$$\ln\left(\frac{y^*}{1 - y^*}\right) = \beta_0 + \sum_{k=1}^{4} s_k(x_k) \tag{1}$$

where $y^*$ is the predicted presence probability, $\beta_0$ is the intercept coefficient, $s$ is the penalized cubic regression spline function to describe the nonlinear environmental effects on the response variable, $x_k$ is the $k$th explanatory variable. We used automatically selected degree of freedom to determine the smoothness of $s$ (*Wood, 2006*). The GAM analysis was performed using the "gam" function of the "mgcv" package in the R platform and the gamma parameter was set to 1.4 to avoid overfitting (*Wood, 2006*, *2014*).

The GWR model is the extension of GLM by accounting for spatial location in the parameter estimates and thus allows for exploring spatially-varying species-environment relationships. The GWR model in this study can be denoted as:

$$\ln\left(\frac{y_i^*}{1 - y_i^*}\right) = \beta_0(u_i, v_i) + \sum_{k=1}^{4} \beta_k(u_i, v_i)x_k \tag{2}$$

where $y_i^*$ is the predicted presence probability at location $i$, $(u_i, v_i)$ is the coordinates of location $i$, $\beta_0$ is the intercept parameter specific to location $i$, $\beta_k$ is the regression parameter for the $k$th environmental variables specific to location $i$. The fixed number of observations (adaptive bandwidth) nearest to the regression point are used to calibrate the local regression models in this study. The weights of observations to local parameter estimates are commonly set as decreasing with the distance to regression point and several forms of function can be used to calculate weights. We used the Gaussian weighting function (Eq. (3)) as its continuity easier for differential calculation.

$$w_{ij} = \exp\left(-\frac{d_{ij}^2}{h}\right) \tag{3}$$

where $d_{ij}$ is the Euclidean distance between the two sample sites $i$ and $j$; $h$ is the bandwidth and has a great impact on the model results. The optimal value of $h$ was selected by minimizing the Akaike's information criterion (AIC). The GWR analysis was performed based on the "GWmodel" package in the R platform (*Gollini et al., 2015*).

The spatial variability of local regression parameter for each environmental variable from the GWR was estimated as the stationary index (SI) (*Brunsdon, Fotheringham & Charlton, 1998*). SI was calculated by dividing the interquartile range of a GWR regression coefficient by twice the s.e. of the same parameter estimate from the global logistic regression model (*Windle et al., 2010*). SI > 1 indicates significant spatial non-stationarity ($p < 0.05$).

The local regression parameter estimates from the GWR for juveniles and adults were interpolated to continuous surfaces and then mapped to visualize spatially-varying environmental effects on the presence probabilities of Yellow Perch. In order to characterize the special zones of species-environment relationships, the local regression coefficient estimates from the GWR were separated into different groups using a *k*-means cluster analysis method. The number of clusters (*k*) was set a prior to 3 and 4 for comparison with basins and MUs respectively. Furthermore, the best number of clusters was estimated based on a gap statistic (*Tibshirani, Walther & Hastie, 2001*). The spatial distribution of clusters was mapped. All the maps were produced by the ArcGIS (Esri, v. 10.2; Redlands, CA, USA) software.

Spatially-varying species-environment relationships were accounted for by pooling the long-term (27 years) survey data to conduct the GWR analysis. However, the presence of Yellow Perch and ecological relationships are likely varying during the survey time. We plotted the GWR residuals by year in order to look for signatures of temporal effects.

## Model evaluation and comparison

Akaike's information criterion and deviance explained (%) were calculated to assess goodness-of-fit for each model. The model with the lower AIC and higher deviance explained would be judged to have better fitting performance. Modelling the binary data can be treated as classified algorithm and a larger area under the receiver operating characteristic curve (AUC) value indicated the higher discrimination accuracy (*Bradley, 1997*). To evaluate whether a model captured the spatial patterns in the response variable, we calculated Moran's I to test for the spatial autocorrelation in model residuals. Value of Moran's I close to −1 and 1 indicates strong clustering and dispersing respectively. A permutation test for Moran's I statistic was used to test for significance of spatial autocorrelation (*Bivand & Wong, 2018*).

To assess the predicted accuracy of the model, the survey data were split into training and testing data randomly as a ratio of 3:1. The training data were used to fit the model and the testing data were used to validate the model. AUC was used to assess the discrepancy between the predicted and observed values. The cross-validation was repeated 100 times for calculating the mean AUC value and its 95% confidence interval.

## RESULTS

Environmental conditions vary widely among the three basins in Lake Erie. Water temperature and depth increase and decrease from east to west respectively (Figs. 1A and 1B). The waters of the east basin are clearer than those in the west and central basins (Fig. 1C). DO concentration does not show marked spatial difference in the Lake Erie (Fig. 1D), primarily because of the post-turnover timing of the central basin survey.

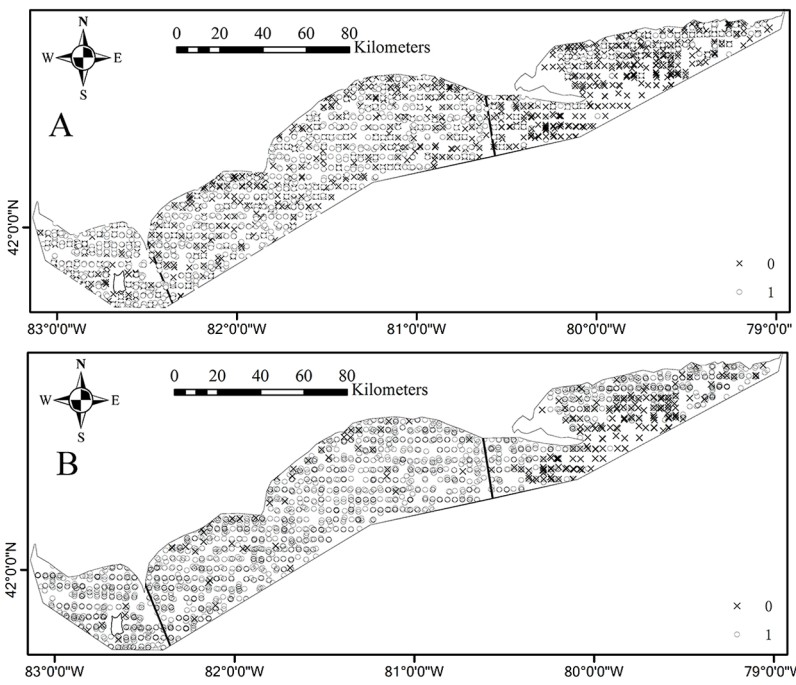

**Figure 2 Spatial distributions of absence (×) and presence (○) for (A) juvenile and (B) adult Yellow Perch in the Canadian side of Lake Erie based on the partnership index survey (PIS) data.**

Juveniles are present at 58% of sample sites, while adults are present at 90% of sample sites. The spatial distribution map indicates that juveniles are mainly distributed in the central and west basins and the high absence is found in the east basin (Fig. 2A). Adults are present in most sample areas and high absences are found in the deep waters of the east basin and near-shore areas in the central basin (Fig. 2B).

Generalized additive model results show that water temperature, water depth, water clarity and DO have significant effects on the presence probability of juveniles, yet only the first three variables significantly affect the adults distribution (Table 1). The presence probability of juveniles significantly increases with water temperature and DO, decreases with water clarity, and first increases and then decreases with water depth (Figs. 3A–3D). The presence probability of adults shows similar change trend with that of juveniles to the variation of water temperature, water depth and DO (Figs. 3E–3G).

Based on the AIC criteria, GWRs with adaptive bandwidths of 64 and 241 points have the best performance for juveniles and adults, respectively. Although the bandwidth was estimated based on model fitting, it indicated the scale of similarity in environment-distribution relationships. GWRs result in marked decreases of AIC values and increases of deviance explained indicating better goodness-of-fit compared with the equivalent GAMs. GWRs also present the high prediction accuracy indicating by the higher AUC values than the equivalent GAMs. Moran's I test results show that spatial autocorrelations of model residuals from the GAMs and GWRs are not significant, implying the two types of models can capture the spatial patterns of the response variable (Table 2).

**Table 1 Test statistics and significance of the four environmental variables used in the generalized additive models (GAMs) for juveniles (GAM-J) and adults (GAM-A).**

| Variables | GAM-J | | GAM-A | |
|---|---|---|---|---|
| | $\lambda^2$ | *p*-value | $\lambda^2$ | *p*-value |
| Temperature | 82.95 | <0.001 | 48.20 | <0.001 |
| Depth | 87.25 | <0.001 | 81.73 | <0.001 |
| Transparency | 28.77 | <0.001 | 46.93 | <0.001 |
| DO | 8.28 | 0.008 | 0.00 | 0.35 |

**Note:**
A variable is significant at $p < 0.05$ in the GAM. J denotes juveniles, A denotes adults, DO denotes dissolved oxygen.

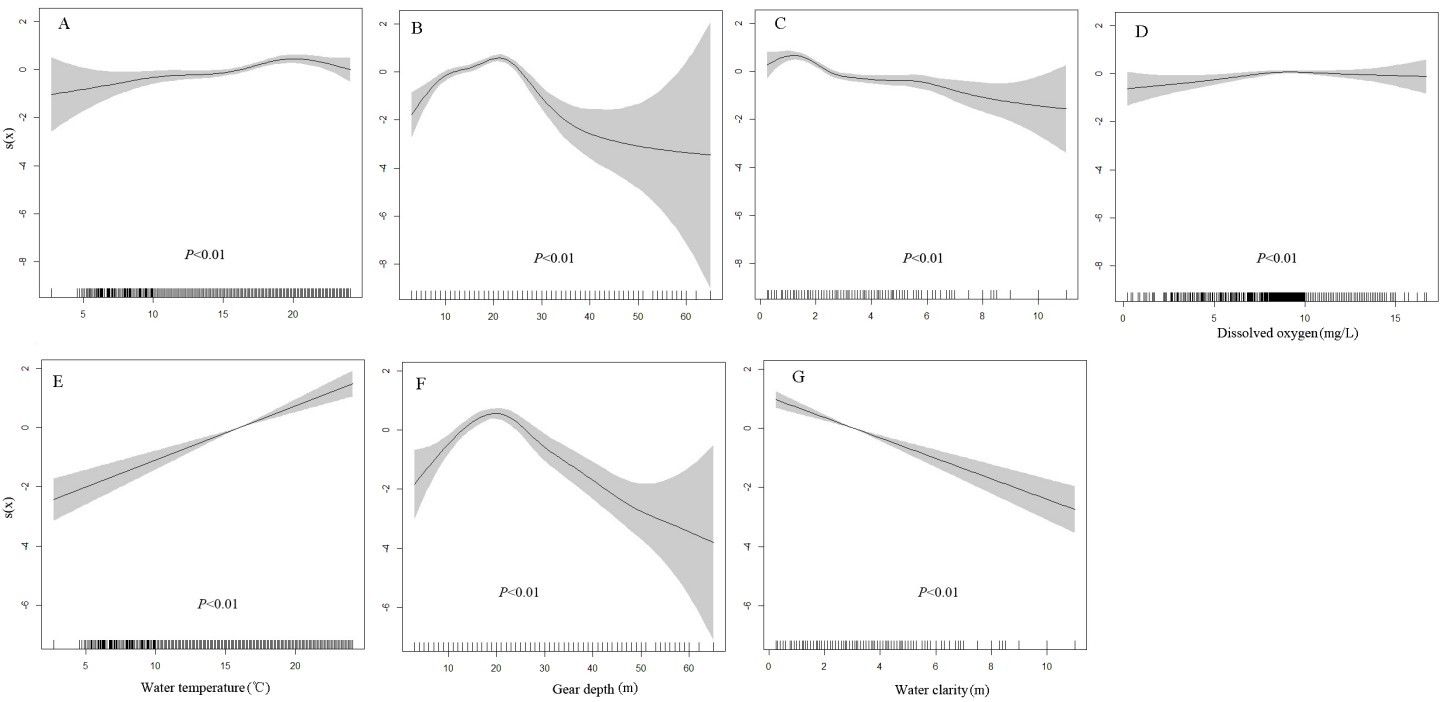

**Figure 3 Environmental effects on the presence probabilities of juveniles and adults based on the generalized additive models (GAMs).**
(A–D) Water temperature, water depth, water clarity and dissolved oxygen on the presence probabilities of juveniles. (E–G) Water temperature, water depth and water clarity on the presence probabilities of adults. Tick marks on the *x*-axis are the observed values for each environment variable; $s(x)$ is the cubic spline function indicating the magnitude and direction of each environment variable on the presence probabilities of juveniles and adults; and shaded areas indicate 95% confidence bounds. The relationship between dissolved oxygen concentration and presence probability of adult was not plotted because it was not significant ($p = 0.35$).

Descriptive statistics of local regression coefficient estimates from the GWRs reveal the much variations of coefficient values. SI values are all greater than 1 indicating the significantly spatial nonstationary relationships between the presence probability of Yellow Perch and environmental variables ($p < 0.05$; Table 3). The estimated coefficient values of water temperature, water depth, water clarity and DO for juveniles from the GWR vary between −0.40 to 0.28, −0.16 to 0.47, −0.86 to 0.25 and −0.51 to 0.55, respectively (Fig. 4). The significant positive associations ($p < 0.05$) between water temperature and presence of juveniles were found in most areas of the east basin and significant

**Table 2 Summary of optimal bandwidths and model performances for generalized additive models (GAMs) and geographically weighted regression (GWR) models.**

| Model | Bandwidth | AIC | Deviance (%) | AUC | CV_AUC ± SD | Moran test $p$-value |
|-------|-----------|-----|--------------|-----|-------------|----------------------|
| GAM-J | – | 2,955.6 | 14.3 | 0.73 | 0.72 ± 0.02 | 0.18 |
| GAM-A | – | 1,018.8 | 36.2 | 0.88 | 0.74 ± 0.02 | 0.51 |
| GWR-J | 64 | 2,809.9 | 23.2 | 0.80 | 0.81 ± 0.01 | 0.96 |
| GWR-A | 241 | 982.3 | 41.5 | 0.91 | 0.90 ± 0.02 | 0.86 |

Note:
The unit of bandwidth is the number of points. J denotes juveniles, A denotes adults. AIC is Akaike's information criterion. AUC is area under the receiver operating characteristic (ROC) curve. CV_AUC ± SD is the mean AUC ± standard deviance calculated based the 100 repeated cross-validations. Moran test is the $p$-values of testing for the significance of residual spatial autocorrelations.

**Table 3 Summary statistics of the logistic GWR local parameter estimates and spatial stationarity index (SI).**

| Model | Variable | Minimum | Lower quartile | Median | Upper quartile | Maximum | SI |
|-------|----------|---------|----------------|--------|----------------|---------|-----|
| GWR-J | Intercept | −9.37 | −2.47 | −0.27 | 1.40 | 6.09 | 4.68 |
| | Temperature | −0.40 | −0.06 | 0.04 | 0.10 | 0.28 | 4.86 |
| | Depth | −0.16 | 0.00 | 0.06 | 0.08 | 0.12 | 8.23 |
| | Transparency | −0.86 | −0.23 | −0.09 | 0.05 | 0.25 | 5.67 |
| | DO | −0.51 | −0.11 | −0.02 | 0.12 | 0.55 | 5.59 |
| GWR-A | Intercept | −10.34 | −4.57 | −0.70 | 3.64 | 5.64 | 5.76 |
| | Temperature | −0.04 | 0.09 | 0.14 | 0.40 | 0.69 | 3.70 |
| | Depth | −0.15 | −0.08 | 0.06 | 0.18 | 0.33 | 11.01 |
| | Transparency | −1.01 | −0.63 | −0.36 | −0.26 | 0.03 | 5.73 |
| | DO | −0.12 | −0.07 | 0.01 | 0.22 | 0.41 | 4.13 |

Note:
SI was calculated by dividing the interquartile range of a GWR regression coefficient by twice the s.e. of the same parameter estimate from the global logistic regression model. SI > 1 indicates significant spatial non-stationarity ($p < 0.05$). J denotes juveniles, A denotes adults, DO denotes dissolved oxygen.

negative associations ($p < 0.05$) were found in the west end of west basin. Two small areas show significant positive associations ($p < 0.05$) and one small area shows significant negative associations ($p < 0.05$) between water temperature and presence of juveniles in the central basin (Fig. 4A). The presence of juveniles presents significant positive associations ($p < 0.05$) with water depth in the west basin and in the shallower waters of central and east basins and significant negative associations ($p < 0.05$) in the deeper waters of east basin (Fig. 4B). Water clarity is negatively related with the presence of juveniles in most areas of Lake Erie and the significant relationships ($p < 0.05$) were found in the central parts of east and west basins and in the east part of central basin (Fig. 4C). Significant positive associations ($p < 0.05$) between the presence of juveniles and DO concentrations were found in the west basin and significant negative associations ($p < 0.05$) were found in some areas of central basin (Fig. 4D). The estimated coefficient values of water temperature, water depth, water clarity and DO for adults from the GWR vary between −0.038 to 0.69, −0.15 to 0.33, −1.0 to 0.027 and −0.12 to 0.41, respectively (Fig. 5).

Peer J

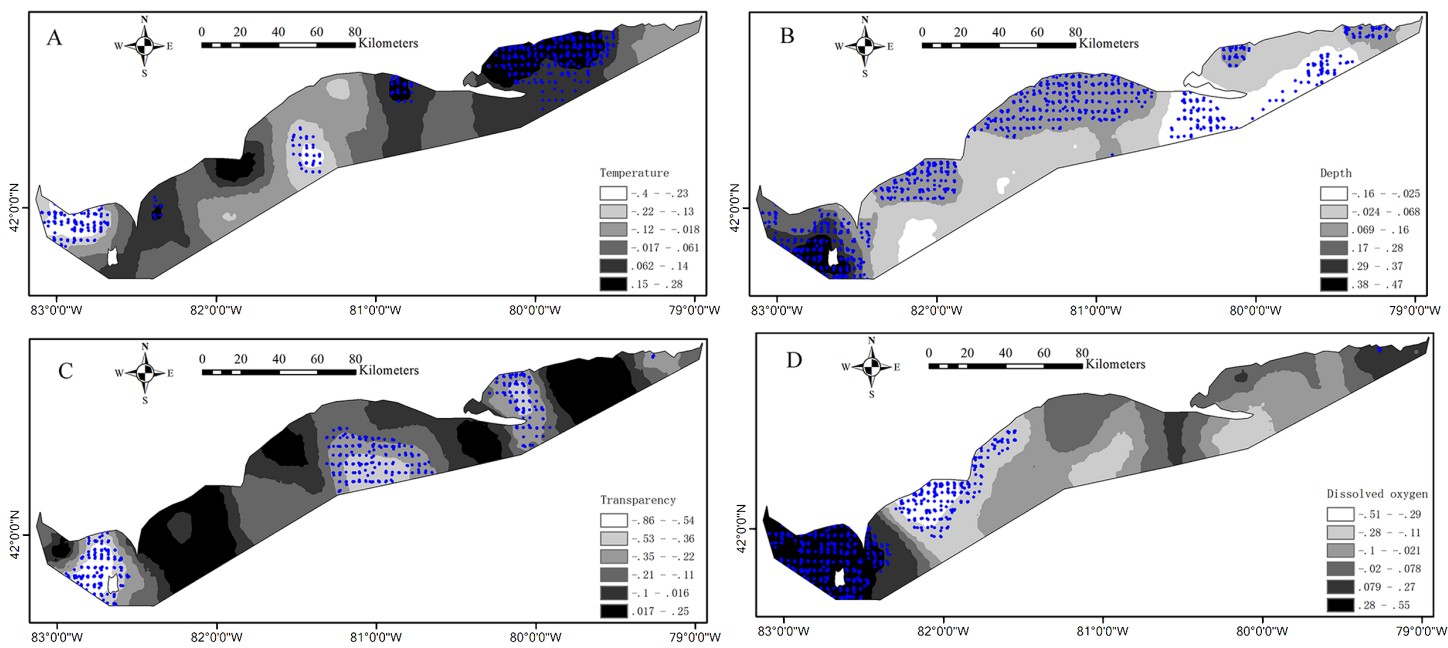

**Figure 4 The interpolated continuous surfaces of the GWR local regression coefficient estimates for juveniles for (A) water temperature, (B) water depth, (C) water transparency and (D) dissolved oxygen.** The blue circles denote that the environment effect was significant ($p < 0.05$) at these sample sites.

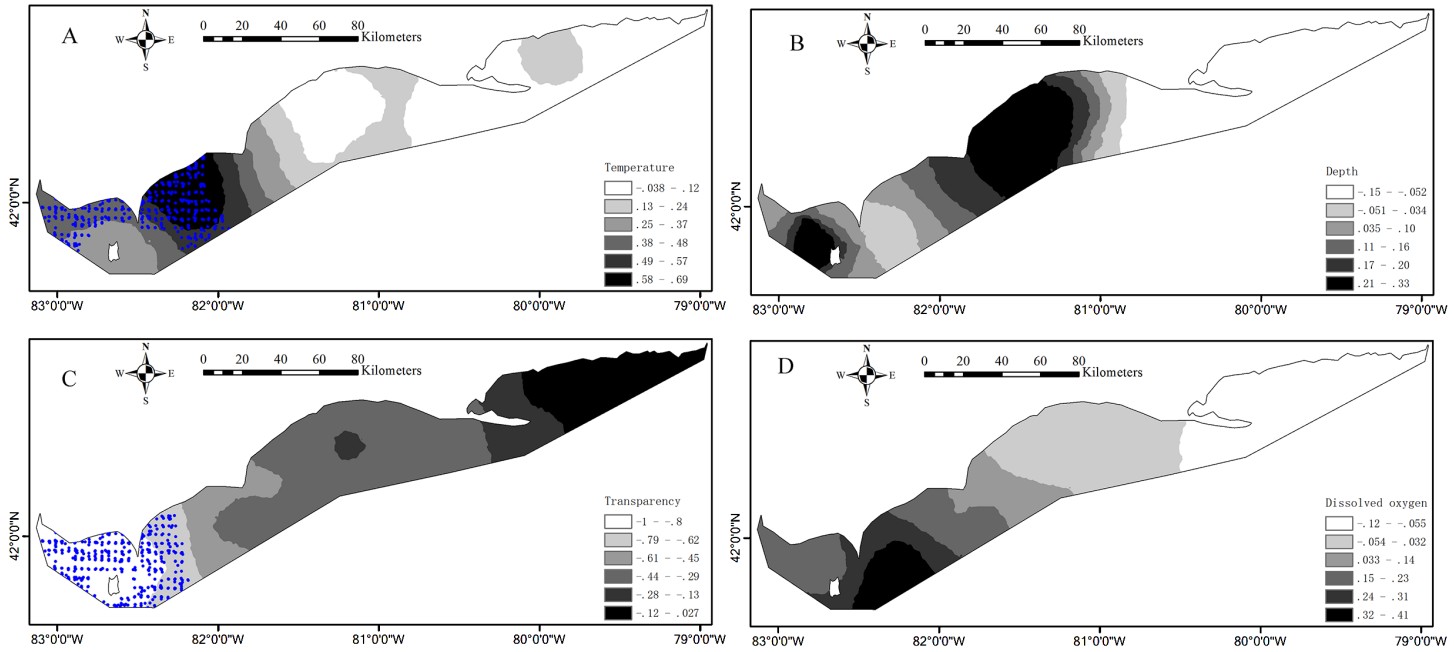

**Figure 5 The interpolated continuous surfaces of the GWR local regression coefficient estimates for adults for (A) water temperature, (B) water depth, (C) water transparency and (D) dissolved oxygen.** The blue circles denote that the environment effect was significant ($p < 0.05$) at these sample sites.

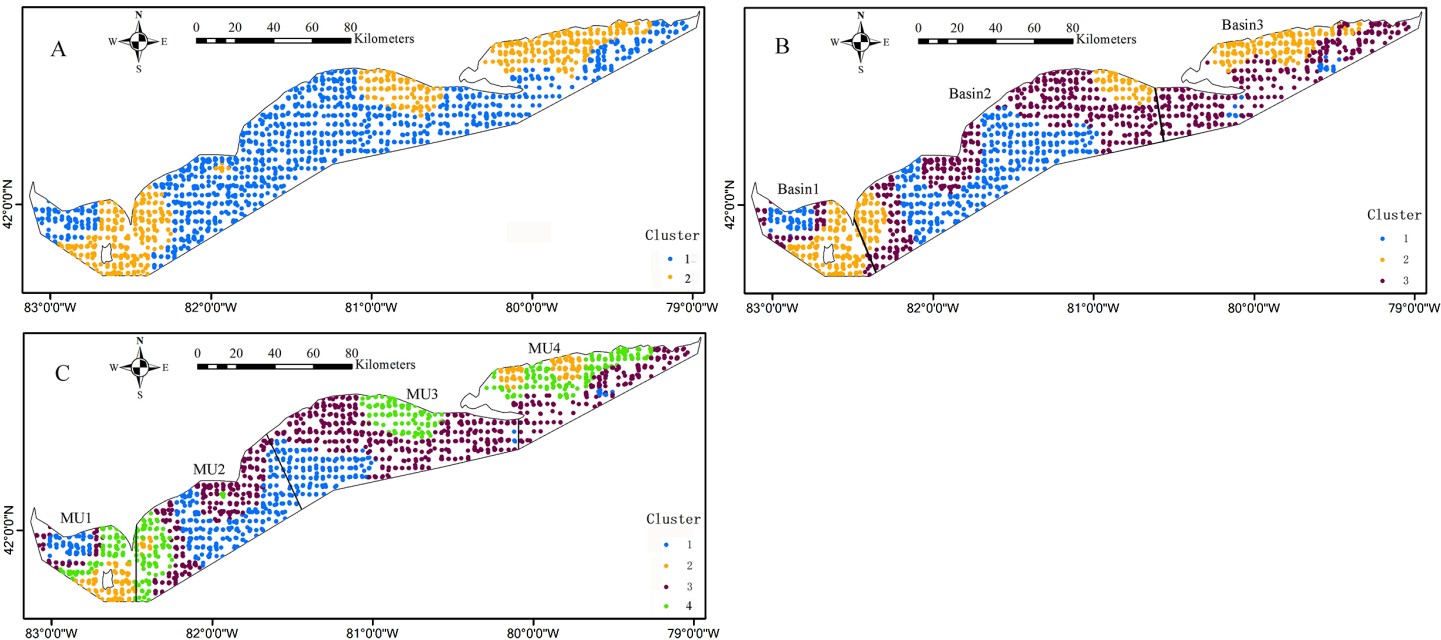

**Figure 6** Mapped results of *k*-means cluster analyses to the local coefficient estimates from the logistic GWR for juveniles, for three clusters, (A) *k* = 2, (B) *k* = 3, (C) *k* = 4. The bold black lines in (B) and (C) are the separate lines among basins and management units, respectively.

The presence of adults increases with water temperature in almost all areas and significant associations ($p < 0.05$) between them were found in the western basin (Fig. 5A). Water depth had no significant effect on the presence of adults for all the sample sites in Lake Erie (Fig. 5B). The negative associations between water clarity and presence of adults are present in almost all areas and the significant associations ($p < 0.05$) were found in the west part of Lake Erie (Fig. 5C). The associations of DO with the presence of adults are not significant for all the sample sites (Fig. 5D).

The *k*-means cluster analysis of local regression coefficient estimates from the GWRs characterized the special zones of environmental effects on the Yellow Perch distribution (Figs. 6 and 7). The *k*-means cluster analysis when *k* = 2 indicates the species-environment relationships for juveniles in the nearshore areas of east basin and part areas of west and central basins are specialized as cluster 2 and the rest of Lake Erie is specialized as cluster 1 (Fig. 6A). As *k* changed from two to three, the areas of cluster 2 do not change and the areas of cluster 1 are further divided into two groups. The different species-environment relationships are found in all the three basins (Fig. 6B). As *k* changed from three to four, the areas of cluster 1 and 3 change little and the areas of cluster 2 are further divided into two groups. The ecological relationships for juveniles in each MU are not classified as one group (Fig. 6C). The *k*-means cluster analysis of *k* = 2 divides Lake Erie into distinct longitudinal zones of environmental effects on the adult distribution (Fig. 7A). The areas of cluster 1 are further cut into two parts as *k* changed from two to three. West and east basins show consistent species-environment relationships (Fig. 7B). As *k* changed from three to four, the east areas (cluster 2) are further separated into

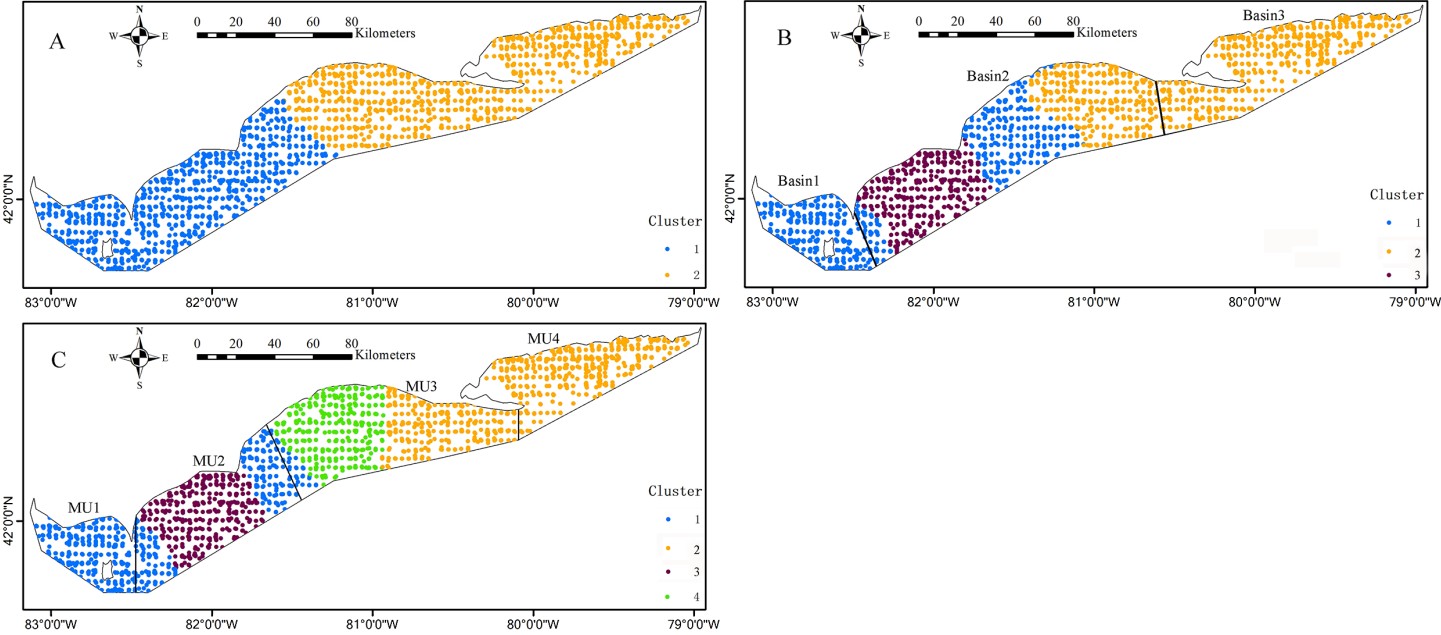

**Figure 7 Mapped results of *k*-means cluster analyses to the local coefficient estimates from the logistic GWR for adults, for three clusters, (A) *k* = 2, (B) *k* = 3, (C) *k* = 4.** The bold black lines in (B) and (C) are the separate lines among basins and management units, respectively.

two adjacent parts and the two groups located in the west of Lake Erie change little. Consistent species-environment relationships were found in MU1 and MU4 (Fig. 7C). Based on the gap statistics, the best numbers of clusters for juveniles and adults are both two.

The GWR residuals changed significantly among years for juveniles ($p < 0.05$). The average residuals markedly lower than zero in 1998, 2001, 2003 and 2005 for juveniles indicated GWR overestimates the probabilities of juveniles in these years. The GWR residuals changed insignificantly among years for adults ($p > 0.05$; Fig. 8).

## DISCUSSIONS

Water temperature is an essential factor for the growth of juvenile Yellow Perch. Juveniles prefer to live in the warmer waters when the water temperature below the optimal range (20.0–23.3 °C) (*McCauley & Read, 1973*; *Power & Heuvel, 1999*). GWR results indicated the presence of juveniles increased with water temperature in the east basin featured with the low water temperature and this was consistent with the previous finding. By contrast, GWR results also proved that the presence of juveniles decreased with water temperature when it over the optimal range (20.0–23.3 °C) for juveniles in the west basin (*McCauley & Read, 1973*). GAM pooled all the survey data and yielded a mean trend in the association of the presence of juveniles with water temperature, which masked the interaction between water temperature and juvenile distribution at finer scales.

Water depth and water clarity in Lake Erie generally increase from west to east. According to the GAM results, the presence of Yellow Perch first increased and then decreased with water depths greater than 20 m. This result projected to space by GWR

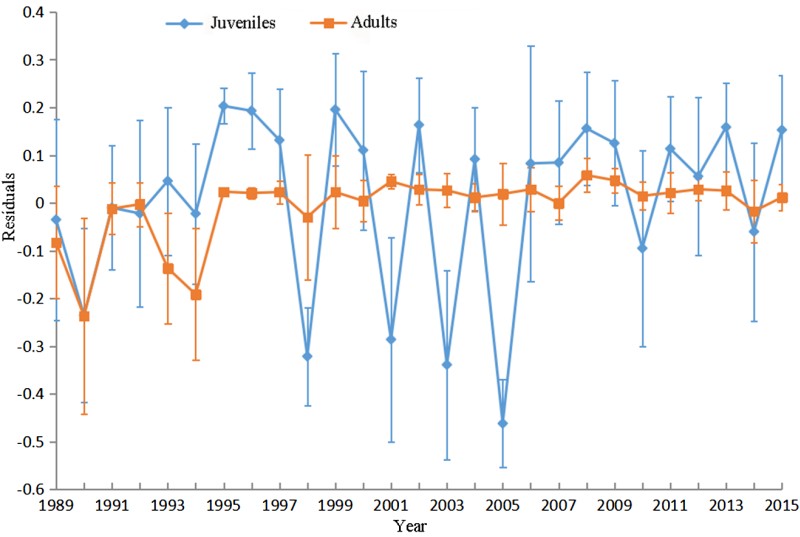

**Figure 8 Mean and standard deviation (SD) values of GWR residuals against year for juveniles and adults.** The filled rectangles and diamonds denote the mean residuals in each year for juveniles and adults respectively and the vertical lines denote SDs.               

was presented as the increment of the presence of Yellow Perch with water depth in the shallower waters and the decrement with water depth in the deeper waters. Juveniles prefer to inhabit the shallower, more turbid waters for avoiding pelagic, visual predators (*Manning et al., 2013*). This finding was verified by the GAM results of the significant decrease of Yellow Perch presence with increasing water clarity. However, clearer waters are good for the growth of juveniles by improving the visual field and increasing the foraging success rates (*Manning et al., 2013*), possibly explaining why the presence of juveniles increased with the water clarity in parts of Lake Erie based on the GWR results.

Dissolved oxygen concentrations below threshold or fluctuating diurnally are not conducive to the growth of juveniles (*Bejda, Phelan & Studholme, 1992*). Hypolimnetic hypoxia (<2 mg $O_2/L^{-1}$) can cause Yellow Perch to escape from the hypoxic habitats to the more oxygenated areas and alter the fish distribution (*Roberts et al., 2012*). Over 99% of the sample sites have DO concentrations above the hypoxic threshold and this is probably the reason to cause the insignificant effect of DO on the presence of adults. *Liu et al. (2018)* also found DO did not affect adult Yellow Perch distribution significantly. GAM results indicated the general trend of the presence of juveniles significantly increasing with DO concentration. This finding may not be appropriate for applying at the local scale. Juveniles prefer to live in the more oxygenated areas for optimizing the growth in the shallower, warmer waters. However, as DO concentration over a certain value, it is not an important factor to affect juvenile's distribution. *Liu et al. (2018)* found the significant interactive effect of DO with water depth on the distribution of juvenile Yellow Perch in Lake Erie. GWR results proved that the effect of DO on juvenile's distribution depending on water depth and are consistent with the earlier findings.

The presence probability of species can be affected by catchability in addition to habitat variables. We assumed the constant catchability over space and time and this may cause bias
to the model results. For example, as a visual forager, adult Yellow Perch may prefer to live in the clearer waters for optimal predation, yet they may detect and avoid gillnets in these waters. The long-term (>20 years) dataset, stratified random sampling design and randomly ordered multifilament mesh sizes of the PIS are expected to reduce any such bias.

Our cluster analysis characterized special zones of species-environment relationships. *Liu et al. (2017)* achieved similar results in analyzing the relationships between walleye distribution and environmental factors in Lake Erie. In order to detect whether a consistent species-environment relationship exists in each basin, we divided the local regression coefficients of GWR into three groups based on the $k$-means cluster analysis. Different ecological relationships were found for juveniles in all the three basins. The sensitivity to spatial environmental variation for juveniles is probably the reason for the different relationships in all the three basins. Consistent ecological relationships were found in the west and east basins for adults. The distinctive environmental attributes with warmer, shallower, more turbid and colder, deeper, clearer waters in the west and east basins respectively may be the reasons to shape the special zones of ecological relationships for adults. Lake Erie was partitioned into four MUs and total allowable catch of Yellow Perch was allocated based on MUs each year (*YPTG (Yellow Perch Task Group), 2015*). The MU boundaries were identified with full consideration of socioeconomic concerns (e.g., at least one major port exists within each MU) and political boundaries (e.g., counties in Ontario) (*Kocovsky & Knight, 2012*). Hence, MUs are convenient for landing and reporting of harvest and may lack of ecological significance to some degree. When comparing the $k$-means cluster analysis ($k = 4$) results for adults with MUs, consistent species-environment relationships were found in MU1 and MU4 and two different kinds of species-environment relationships were found in MU2 and MU3. This implied the variation of species-environment relationships at a scale finer than the MU. *Henderson & Nepszy (1989)* reported central basin Yellow Perch were larger at age than western basin Yellow Perch. They also noted the different mortality and growth rates. *Sepulveda-Villet & Stepien (2011)* and *Kocovsky & Knight (2012)* provided strong evidence based on genetic and morphological analysis respectively that Yellow Perch exist as discrete stocks in Lake Erie and that stocks exist at scales finer than the current MUs. Our analysis further facilitates verification of stocks for Yellow Perch from the perspective of species-environment interactions. Management of Yellow Perch should benefit through considering such kind of species-environment differentiations. Comprehensive analysis with additional explanatory variable included in the GWR coupled with genetic and morphological research can provide supports for identification of unique stocks and refinement of the current MU structure in consideration of ecological relevance for sustainable management of Yellow Perch.

Density data (catch weight) were available for Yellow Perch at each sample site. Adults were present at 90% of sample sites and the absences were mainly distributed in the east basin. Analysis based on the presence/absence data of adults may mask the species-environment relationships at finer scales because of the evident low proportion of absence data. The difference of density data for adults was significant among sample sites ($p < 0.05$). Further analysis based on density is suggested in the future.

The predominant advantage of GWR is the ability to capture the spatially-varying ecological relationships. Furthermore, GWR can be used as an identifier to determine at which scale the species-environment relationships become stationary (*Windle et al., 2010*). Generalized additive mixed model (GAMM) is also able to allow the spatial variation to be described using a complex spatial field, such as a Gaussian random field. Due to the complexity of GAMM, it has seldom been used to explore spatial nonstationary species-environment relationships. For comparison with GWRs, we also used GAMMs to estimate spatially-varying ecological relationships. The performance of GAMMs was improved over that of GAMs as indicated by the lower AIC values, but GWRs were still superior to GAMMs. *Liu et al. (2017)* also found the superiority of GWR over GAMM when spatially nonstationary species-environment relationships were present. The main advantages of GWR over GAMM are that we can map the local relationships from the GWR and identify the distinct zones of species-environment relationships.

The presence and response of Yellow perch to environmental variations may vary among years. The long-term (27 years) survey data were pooled and the temporal effects were not accounted for when conducting GWR analysis in this study. This resulted in lower goodness-of-fit and higher model residuals in several years. For example, GWR provides inferior performance in 1998, 2001, 2003 and 2005 because the presences of juveniles in these years were markedly lower than the mean presence calculated based on the data from all years. Multi-level categorical variable (e.g., year in this study) was not recommended as an explanatory variable in the GWR analysis because of the strong risk to cause collinearity in the local regression coefficients (*Wheeler & Tiefelsdorf, 2005*). Geographically and temporally weighted regression (GTWR) was developed by incorporating temporal effects into the GWR to deal with both spatial and temporal nonstationarity simultaneously (*Fotheringham, Crespo & Yao, 2015*). Future research in this area should focus on estimating spatial and temporal nonstationary ecological relationships for Yellow Perch in Lake Erie.

Although the GWR was shown to be superior to the global regression models, it should be used with caution. Due to local regression coefficients estimated based on the neighborhood observations, GWR cannot be used to predict species distribution outside the study area. Spatial coordinates are the only information required by GWR to estimate local regression coefficients at unobserved locations. Thus, GWR cannot be used to predict future distribution of species. The possible collinearity in local regression coefficients may limit the interpretation of species-environment relationships (*Wheeler & Tiefelsdorf, 2005*). The prediction accuracy of GWR is sensitive to data quantity. Thus, conducting the GWR separately for each year in this study may not be sufficient to get ecologically meaningful results. The large data quantity required to estimate local regression coefficients also limits the application of GWR.

## CONCLUSIONS

Developing a global regression model by pooling all the survey data in the large region may mask the local variability in the processes being studied although such an approach is more convenient to conduct. We applied the GWR to test the assumption of spatial

stationarity in estimating the relationships between Yellow Perch distribution and environmental variables in Lake Erie. Our analysis demonstrated that environmental effects on the Yellow Perch distribution varied significantly among locations, and verified the existence of discrete species-environment relationships for Yellow Perch in Lake Erie. Our analysis further facilitates verification of discrete Yellow Perch stocks from the perspective of species-environment interactions. Management of Yellow Perch should benefit through considering such species-environment spatial differentiations. Despite its limitations, GWR is recommended as a complementary tool for global regression models in exploring spatially-varying ecological relationships. We suggest expanded research is needed to explore spatio-temporal nonstationary species-environment relationships for Yellow Perch in Lake Erie using a GTWR model.

## ACKNOWLEDGEMENTS

We thank the Ontario Commercial Fisheries' Association, Lake Erie Management Unit of the Ontario Ministry of Natural Resources and Forestry and the Lake Erie Committee of the Great Lakes Fishery Commission for providing the data to us.

### Funding

Sources of support include a grant "Integration of spatial stock structure and multiple stocks into stock assessment for Yellow Perch in Lake Erie" funded by the Ontario Commercial Fisheries' Association to Dr. Yan Jiao at Virginia Tech (fund code #458000), and a scholarship from the Ocean University of China to Dr. Changdong Liu to work in Dr. Jiao's lab. There was no additional external funding received for this study. The funders had no role in study design, data collection and analysis, decision to publish, or preparation of the manuscript.

### Grant Disclosures

The following grant information was disclosed by the authors:
Integration of spatial stock structure and multiple stocks into stock assessment for Yellow Perch in Lake Erie.
Ontario Commercial Fisheries' Association at Virginia Tech: fund code #458000.
Ocean University of China.

### Competing Interests

The authors declare that they have no competing interests.

### Author Contributions

- Changdong Liu conceived and designed the experiments, analyzed the data, contributed reagents/materials/analysis tools, prepared figures and/or tables, authored or reviewed drafts of the paper, approved the final draft.
- Junchao Liu analyzed the data, prepared figures and/or tables, authored or reviewed drafts of the paper, approved the final draft.

- Yan Jiao conceived and designed the experiments, performed the experiments, analyzed the data, contributed reagents/materials/analysis tools, authored or reviewed drafts of the paper, approved the final draft.
- Yanli Tang analyzed the data, prepared figures and/or tables, approved the final draft.
- Kevin B. Reid conceived and designed the experiments, performed the experiments, contributed reagents/materials/analysis tools, authored or reviewed drafts of the paper, approved the final draft.

## Data Availability

The raw measurements are available in the Supplemental Files.

## Supplemental Information

Supplemental information for this article can be found online at http://dx.doi.org/10.7717/peerj.7350#supplemental-information.

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
