# Peer review of "Exploring spatial nonstationary environmental effects on Yellow Perch distribution in Lake Erie"

_PeerJ, doi:10.7717/peerj.7350_

## Round 0.1 · original submission · Major Revisions

Both reviewers have good comments which will help improve your manuscript. In particular, the raw data need to be better described to allow for a full examination of the manuscript. In addition, there appears to be an inconsistency in figure 3 where the plotted occurrences are identical for adult and juvenile fish. Please focus your revision on these and the other relevant comments.

Reviewer 1 ·

Basic reporting

Overall, the article was written in professional English. However, some sentences were not clear and require restructering (line 104; lines 193-194; lines 241-242; lines 285-286; liens 295-296; lines 306-307). Throughout the introduction, methods and discussion ample references are included thereby providing sufficient background/context. A reference for the used package and R statistics is required at line 164 and line 183. The structure of the manuscript conforms to the standard structure. The Figures are relevant and of high enough detail. Readability of figures could be improved by giving the units for the x-axis (Dissolved Oxygen --> mg/l). Based on the description in the text it was impossible to check the raw data for consistency. Headers of raw data did not correspond to the text. Moreover, total number of datapoints mentioned in text (2502) dit not match with the number of datapoints in the raw data (2610). The presented results are sufficient for a publication. The reporting of statistical results is limited, no test values are given. A GAM table with the results for each abiotic factor is missing. At line 222-223 give test values for the statement that a significant difference was found. Can you state that there is a significant difference in AIC at line 235 ?

Experimental design

The manuscript falls within the aims and scope of the journal. The introcution clearly underpins the added value and the research gap of the goal of the manuscript. However, the actual phrasing of the goal can be improved (lines 105-115). It should be formulated more like a goal then as an activity.

Large parts of the methods are described in detail but due to some inconsistenties it does not provide sufficient detail to replicate. The total number of datapoints included mentioned in line 126 does not correspond with the raw data file. Moreover, the headers of the raw data are not self explanatory. Additional information on all headers is required. The two different age groups can not be distinguished in the raw data file. The estimation of fish age mentioned in line 129 requires more explanation. Was this performed by the authors or was this already available in the used dataset. How was this performed? consistent with previous studies? In the methods it is stated that the near bottom living fish were included however the abiotic condtions were mainly measered on the top of the gill net, which differs in space. Potential confounding effects of this depth effect should be mentioned in the discussion. At line 136/137 it is explicetely stating that bottom dwelling fish data was used, does this refer to the previously mentioned 2502 samples or is this a subset?

Validity of the findings

The data could not be checked fully due to unclear headers and discrepancy in the number of datapoints included. Moreover, the methodology was not sufficiently clear regarding fish age. Overall the conclusios based on the performed analyses are clear, linked to the original research question. At line 338-339 a reference is made to the k=4 analyses whereas previously it was indicated that the k=2 analysis provided the best results. Why then discuss the outcomes of the k=4 analysis? The analysis was now limited to presence-absence data, however density data was also collected. The discussion could be improved by discussing not analysing density data.

Figure 3 seems to have an error: the observed datapoints for both the adults and juveniles appear to be identical. This would then inherently results in actually the same relation between presence absence and the environmental factors which is not found. This is especially clear for adult fish in relation to temperature where the density of occurrences does not correspond to the plotted relation.

Reviewer 2 ·

Basic reporting

The manuscript provides a useful exercise to explore the spatial non-stationary environmental effects on Yellow perch in Lake Erie. I think this is a worthy study for PeerJ but I do have several critiques that I think need to be addressed before publication. The critiques are summarized in the general comments.

The main context of the manuscript is clear but it needs to be checked for grammar issues. The overall structure of the manuscript follows the logic. The manuscript can be improved with the expansion of discussion and better labels in some figures (e.g. Figure 1 and Figure 6). The details are explained in the general comments section in this review.

Experimental design

The study is original primary research within the scope of the journal. Such studies could be helpful to the Yellow perch fishery in Lake Erie area and can serve as an example of how to apply the GWR model as a complementary tool for species distribution modeling. The manuscript clearly states the goal of the study and well described the methods with sufficient details, but there is little explanation of the research question, the authors need to further explain why biologically these relationships should vary across this relatively small study area.

Validity of the findings

This study is a fine application of the existing package and previously developed model on the distribution of Yellow perch. The data and methods are robust and statistically sound. However, since this is a case study of Yellow perch distribution, I recommend the authors provide more biological speculation/interpretation of the findings instead of statistical interpretation. The conclusion is clearly stated but it is very similar compared with other studies that used the GWR model. The authors need to provide a better ecological/biological conclusion related to the Yellow perch distribution.

Additional comments

The authors need to provide a better ecological/biological meaning of the relationships found between the presence of Yellow perch and environmental variables as well as the motivation of the study in the introduction. There is little explanation for why biologically these relationships between the presence of Yellow perch and environmental variables should vary across the study area. The study area is a relatively small area and the population here could be well mixed if it is a single population. Given this fact, what is the motivation for allowing spatially varied relationships between environmental variables and the presence of Yellow perch? From the results, we could see the GWR fits the data slightly better than a global model of GAM, but given the biology, do the estimated relationships could vary so rapidly over a short distance? What are the biological implications of the estimated relationships?

Lines 91-93: The authors mentioned that the study area has significant environmental variations in the west, central, and east basins. In the absence of any plots of the abiotic variables themselves, it is difficult to assess the impact of the local environmental conditions on the occurrence of Yellow perch across space. Perhaps provide the observed environmental data maps in the manuscript could help the audience.

Generalized additive models (GAM) is also able to allow the spatial variation to be described using a complex spatial field, such as a Gaussian random field. The authors need to mention the reasons why they chose the GWR and the simple GAM in this study, and discuss the possibility to use GAM with a random field to consider spatial variations in the introduction or discussion.

There are 27 years of data (1989-2015), the presence of Yellow perch are likely varying during that time, but the authors only mentioned in the very end of the discussion that no temporal term was included in the models. I suggest the authors plot the residuals by year to look for signatures of temporal effects. The authors also need to discuss how inter-annual changes in the population presence or different regional environmental changes could affect their results.

Line 158: The authors mentioned that they used a fixed number of observations as bandwidth. Do the bandwidth estimates have a biological interpretation? If so, please provide its implication in the results.

Line 201: The authors mentioned that the juveniles were found at 58% of sample sites, but adults were found at 90% sample sites. This is opposite to the information from Figure 2, which shows a very low occurrence of adults in the study area. It also could be the caption was not right.

The GWR is a moving window of GLM at each study site. Are the estimated coefficients of each variable significant at each site? The authors need to describe the significance of the estimated relationships in the results.

Variations in catchability can be confounded with the variation in the distribution in spatial modeling research. The authors did not discuss the issue of catchability in this paper. I suggest the authors expand their discussion on the assumption of constant catchability over space and time and caveats associated with this assumption.

Table 1: Change 'Moran test' to 'Moran test p-value'.

Table 2: If I understand correctly, the data/variables used in some final models were different (e.g. GAM-Adult with three x variables and GWR-Adult with four x variables), so the AIC values cannot be used to compare the models with different input data. I suggest removing the AIC column in the table.

Figure 1: The survey sites need to be clearly shown on the map with the same size of circles. The information provided from figure 1 could be found in figure 2 as well, so I suggest drawing the points with colors that represent values of observed environmental data (e.g. temperature and depth) in figure 1.

Figure 2: The resolution of the figure needs to be improved. Please double check the caption and labels in the figure. I think the circle represents the presence of species.

Figure 4 and Figure 5: Please provide the lines for separating the basins on the map.

Figure 6: The k-mean cluster analysis was conducted using the t-values from the model outputs. Why not using the coefficient estimates? t-values could be used to compute p-values and help determine whether or not the estimated relationships exist. If the authors want to "characterize the spatial zones of species-environmental relationships" (Lines 178-180), I suggest applying the k-mean cluster analysis to the estimated coefficients which represent the magnitude and direction of estimated relationships.

---

## Round 0.2 · accepted · Accept

Thank you for your efforts in revising your manuscript.

Reviewer 1 ·

Basic reporting

The authors have thoroughly adapted their MS based on the comments of both reviewers. Readability has been improved throuhgout. References have been updated. The structure of the MS is professional and corresponds to the journal.

Experimental design

The experimental design, especially the raw data, has improved and it is now more clear why specific decisions were made. Replication is now possible.

Validity of the findings

With the expansion of the discussion the impact of the findings is now more clear.

Reviewer 3 ·

Basic reporting

The authors have met these criteria.

Experimental design

The authors have met these criteria.

Validity of the findings

The authors have met these criteria.

Additional comments

please see the attachment.

Annotated reviews are not available for download in order to protect the identity of reviewers who chose to remain anonymous.